analytical chemistry/chemical biology

circular dichroism, amino acid, enantiomer, quantification

**Authors for correspondence:**
Jianxi Ying
e-mail: yingjianxi@nbu.edu.cn
Yufen Zhao
e-mail: zhaoyufen@nbu.edu.cn

[†]Present address: Institute of Drug Discovery Technology, Ningbo University, No. 818 Fenghua Road, Ningbo, 315211, China.

This article has been edited by the Royal Society of Chemistry, including the commissioning, peer review process and editorial aspects up to the point of acceptance.

# An electronic circular dichroism spectroscopy method for the quantification of L- and D-amino acids in enantiomeric mixtures

Ruiwen Ding[1,2], Jianxi Ying[1,2,†] and Yufen Zhao[1,2,†]

[1]Institute of Drug Discovery Technology, and [2]Qian Xuesen Collaborative Research Center of Astrochemistry and Space Life Sciences, Ningbo University, No. 818 Fenghua Road, Ningbo, Zhejiang 315211, People's Republic of China

JY, 0000-0002-1035-5345

Rapid quantitative analysis of single chiral amino acid (aa) was achieved using circular dichroism (CD) with data analysis by standard calibration curve. The absolute concentrations of D- and L-aas in enantiomeric mixtures were determined by CD and achiral liquid chromatography (LC) method. It is worth noting that CD and LC were used independently, not online LC/CD in this study. The errors of the experimental results were less than 10%. The method is also applicable to the quantification of non-aa chiral molecules, such as chiral nucleoside and chiral quinine. With this study, we provide a new method for the chiral quantitative analysis of enantiomeric aas mixtures.

## 1. Introduction

Amino acids are the most important chiral biomolecules that act as the building blocks of peptides (proteins) in living organisms [1]. Although L-amino acids (L-aa) are selected as the predominant conformation materials to form the peptides by the natural selection, D-amino acids (D-aa) can also been found in organisms and show important effects on biological processes [2–4]. Analytical methods for chiral amino acid (aa) are of unquestionable and general importance. At present, liquid chromatography (LC), mass spectrometry (MS) and nuclear magnetic resonance (NMR) spectroscopy are usually employed for the analysis of chiral molecules [5–7]. LC identification and quantification of L-/-aas in enantiomeric mixtures can be achieved by chiral column or precolumn derivatization [4,8–13]. The application of MS in chiral analysis is based on

mass-resolve adducts generated by the additives and the analytes [14–17]. Chiral reagents are used for chiral analysis of NMR [18,19].

It is worth noting that although a general method to analyse chiral compounds is based on chiral LC, other methods also display advantages and may complement the chiral analysis techniques. In order to meet the requirement of increasing complex and diverse chiral samples, it is imperative to develop new chiral analysis methods. To the best of our knowledge, the quantitative determination of D, L-aa using circular dichroism (CD) spectroscopy remains unreported. CD spectroscopy is an important analytical technique applied to the enantiomeric excess, conformation and structural analysis for chiral molecules [20–25], especially for proteins and nucleic acids. The CD spectroscopy can also be used in the determination of protein, nucleic acid and small molecule concentration [26–28].

In this study, we present a simple and rapid method for the determination of chiral aa concentration based on CD spectroscopy. For the single chiral aa, the quantification of sample was achieved by standard calibration curve using CD. Meanwhile, the quantification of L- and D-aa in enantiomer mixtures was determined by calibration curves using CD and achiral LC spectroscopy. Besides chiral aa, the quantitative analysis of chiral nucleoside and quinine also showed good correlation between concentration and CD signal intensity, respectively.

# 2. Material and methods

## 2.1. Materials and instruments

L-Proline (L-Pro), L-methionine (L-Met), L-valine (L-Val), L-histidine (L-His), L-isoleucine (L-Ile), L-leucine (L-Leu), L-threonine (L-Thr), L-tryptophan (L-Trp), L-tyrosine (L-Tyr), L-alanine (L-Ala), L-serine (L-Ser), L-phenylalanine (L-Phe), L-arginine (L-Arg), L-asparagine (L-Asn), L-aspartic acid (L-Asp), L-cysteine (L-Cys), L-glutamine (L-Gln), L-glutamic acid (L-Glu) and L-lysine (L-Lys) are from Beijing Solarbio Science & Technology Co., Ltd. D-Proline (D-Pro), D-methionine (D-Met),D-leucine (D-Leu) and D-tryptophan (D-Trp) are from Tokyo Chemical Industry Co., Ltd. D-Guanosine is from Beijing Solarbio Since & Technology Co., Ltd. Quinine is from Saen Chemical Technology (Shanghai) Co., Ltd.

Analytical reagents methanol (HPLC grade) was purchased from Spectrum Chemical, and formic acid was purchased from Sigma Aldrich. Ultrapure water from a Heal Force water purification system (Pudong, Shanghai, China) was used to prepare solutions and the mobile phase.

The aa measurements were performed on Jasco-J1700 electronic CD at ambient temperature. The HPLC was performed on Agilent 1260 Infinity system and fitted with an Agilent TC- C18, 5 µm, 4.6 mm × 150 mm column. The column temperature was maintained at room temperature. The HPLC flow rate was 1 ml min$^{-1}$ with TC-C18.

## 2.2. Quantitative analysis of circular dichroism

CD is an absorption spectroscopy method based on the differential absorption of left and right circularly polarized light in optically active chiral sample. The CD signal ($\Delta A$) of chiral sample can be described by the following equation:

$$\Delta A = A_L - A_R = \Delta \varepsilon \cdot c \cdot l. \tag{2.1}$$

In the above equation, $A_L$ and $A_R$ are the absorbance values of the left circularly polarized (LCP) and right circularly polarized (RCP) light by the chiral sample; $\Delta \varepsilon$ is the molar circular dichroism, an intrinsic property of the chiral sample; $c$ is the molar concentration of the chiral sample; and $l$ is the path length in centimetres (cm). The value of $\Delta \varepsilon$ is a constant at stable conformations of the chiral sample at a given wavelength.

Although $\Delta A$ is usually described, the CD spectrum is often reported in degrees of ellipticity $\theta$ for historical reasons. And the $\Delta A$ and $\theta$ are readily interconverted by the following equation:

$$\theta = 3298 \cdot \Delta A = 3298 \cdot \Delta \varepsilon \cdot c \cdot l, \tag{2.2}$$

where the parameter 3298 converts $\Delta A$ from the units of molar circular dichroism to historical units of degrees $\cdot$ cm$^2 \cdot$ dmol$^{-1}$.

Then the equation can be written by defining $c$ as

$$c = \frac{\theta}{3298 \cdot \Delta\varepsilon \cdot l}. \tag{2.3}$$

Therefore, $c$ is directly proportional to $\theta$ within a certain concentration range. As we know, the LC spectroscopy is usually applied to quantifying the concentration of aa based on a standard calibration curve. One of the requirements for LC quantification is the sample has absorption of UV–vis. CD is also an absorption spectrometry. By the same token, we expect to determine the aa concentration based on equation (2.3) using CD spectroscopy.

## 2.3. Quantify the concentration of amino acid

Dissolve L-/D-aa in pure water to make mother liquors, which are then diluted to five different concentrations and tested by CD. Make calibration curves based on CD test results.

## 2.4. Quantify the concentration of L-/D-amino acid in enantiomeric mixture

The CD signal of L-aa and D-aa can be expressed by applying equation (2.3) as follows:

$$c_1 = \frac{\theta_L}{3298 \cdot \Delta\varepsilon \cdot l} \tag{2.4}$$

and

$$c_2 = \frac{\theta_D}{3298 \cdot \Delta\varepsilon \cdot l}, \tag{2.5}$$

where $c_1$ and $c_2$, respectively, are the concentration of the L-aa and D-aa. Since L-aa gave positive cotton effects and D-aa gave negative cotton effects at 190–220 nm, one of the values of $\theta_L$ and $\theta_D$ is positive and the other is negative.

Then

$$n_a = |n_1 - n_2| \Rightarrow c_aV = |c_1V - c_2V| \Rightarrow c_aV = |c_1 - c_2|V \Rightarrow c_a = |c_1 - c_2|, \tag{2.6}$$

where $n_1$ and $n_2$, respectively, are the amount of substance of the L-aa and D-aa, $n_a$ is the absolute difference of $n_1$ and $n_2$. $c_1$ and $c_2$, respectively, are the concentration of the L-aa and D-aa, $c_a$ is the absolute difference of $c_1$ and $c_2$. $c_a$ is obtained from the calibration curve of CD signal.

It is well known that the LC method usually applied to determination of the concentrations of aa is based on a standard calibration curve of the UV–vis absorbance. The peaks of the enantiomer mixtures in the achiral LC are perfectly coincident. So the concentration $c_b$ of enantiomer mixtures obtained through the calibration curve of LC signal can be described by the following equation:

$$n_b = n_1 + n_2 \Rightarrow c_bV = c_1V + c_2V \Rightarrow c_bV = (c_1 + c_2)V \Rightarrow c_b = c_1 + c_2, \tag{2.7}$$

where $n_1$ and $n_2$, respectively, are the amount of substance of the L-aa and D-aa, $n_b$ is the sum of $n_1$ and $n_2$. $c_1$ and $c_2$, respectively, are the concentration of the L-aa and D-aa, $c_b$ is the sum of $c_1$ and $c_2$. $c_b$ is obtained from the calibration curve of LC signal.

Combining the last two equations (equations (2.6) and (2.7)), values of $c_1$ and $c_2$ can be obtained by simple linear equations computing.

# 3. Results and discussion

## 3.1. Quantification of single chiral amino acid

To test the possibility of using CD spectroscopy to quantify the concentration of single aa, a series of known concentrations of 19 natural L-aa were measured by CD (figure 1), and construction of calibration curves involves five points of the corresponding pure L-aa (figure 2; electronic supplementary material, figures S1–S16). The coefficient of determination ($R^2$) is at least 0.999 for curves. The results showed that linear correlation between CD signal $\theta$ and L-aa concentration $c$ was found in a certain concentration range.

Then, these curves were used for direct comparison of the relative peak signal intensity of the corresponding known concentration of four L-aa (L-Leu, L-Pro, L-Met and L-Trp) samples. Table 1

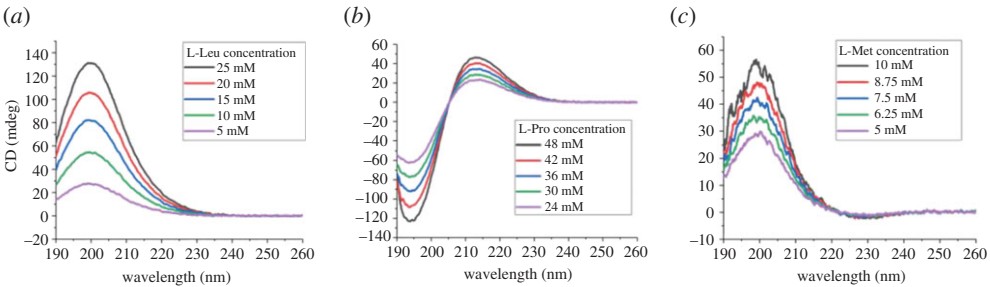

**Figure 1.** CD spectra of three L-amino acids. (*a*) Various concentrations of L-Leu; (*b*) various concentrations of L-Pro; (*c*) various concentrations of L-Met. All the experiments data reported here were repeated three times.

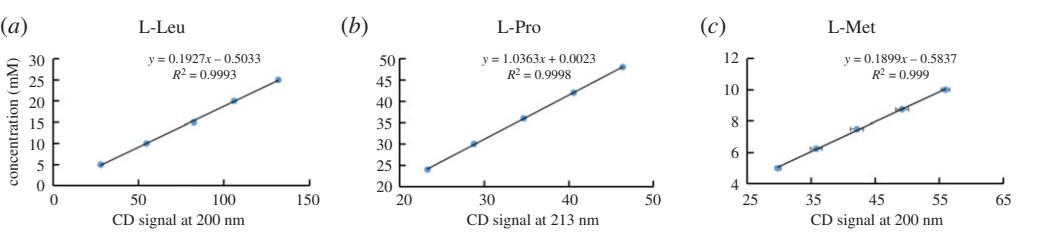

**Figure 2.** The construction of calibration curves for L-amino acids using CD spectra. (*a*) Calibration curve of the L-Leu; (*b*) calibration curve of the L-Pro; (*c*) calibration curve of the L-Met. All the experiments data reported here were repeated three times.

**Table 1.** Accurate CD calibration curves measurements of amino acids. $\Delta$ *diff.* $= |Exp. - Cal.|/Cal. * 100\%$. All of *difference* are less than 8%. All the experiments data reported here were repeated three times.

| aa[a] | | L-aa[b] | | | D-aa[c] | | |
|---|---|---|---|---|---|---|---|
| Leu (mM) | *Cal.*[d] | 8.00 | 13.00 | 18.00 | 8.00 | 13.00 | 18.00 |
| | *Exp.*[e] | 8.00 | 14.00 | 19.00 | 8.00 | 14.00 | 19.00 |
| | $\Delta$ *diff.*[f] | 0 | 7.7% | 5.6% | 0 | 7.7% | 5.6% |
| Pro (mM) | *Cal.*[d] | 40.00 | 25.00 | 32.00 | 40.00 | 25.00 | 32.00 |
| | *Exp.*[e] | 40.00 | 24.00 | 31.00 | 40.00 | 26.00 | 33.00 |
| | $\Delta$ *diff.*[f] | 0 | 4.2% | 3.1% | 0 | 4.0% | 3.1% |
| Met (mM) | *Cal.*[d] | 4.00 | 6.00 | 7.00 | 4.00 | 6.00 | 7.00 |
| | *Exp.*[e] | 4.00 | 6.00 | 7.00 | 4.00 | 6.00 | 7.00 |
| | $\Delta$ *diff.*[f] | 0 | 0 | 0 | 0 | 0 | 0 |
| Trp (mM) | *Cal.*[d] | 0.20 | 0.40 | 0.55 | 0.20 | 0.40 | 0.55 |
| | *Exp.*[e] | 0.20 | 0.40 | 0.52 | 0.20 | 0.40 | 0.55 |
| | $\Delta$ *diff.*[f] | 0 | 0 | 5.5% | 0 | 0 | 0 |

[a]Amino acid.
[b]L-amino acid.
[c]D-amino acid.
[d]The calculated value of the amino acid.
[e]The experimental value of the amino acid.
[f]The relative error between the calculated value and the experimental value.

lists the calculated concentration of samples and experimental values obtained through the calibration curves. Meanwhile, similar test with four D-aa (D-Leu, D-Pro, D-Met and D-Trp) samples had also led to the good correlation (figure 3; electronic supplementary material figures S17–S20). The differences in percentage between the experimental value and the calculated concentration were between 0 and 7.7 (table 1). The concentration range of standard solution analysis for each sample is

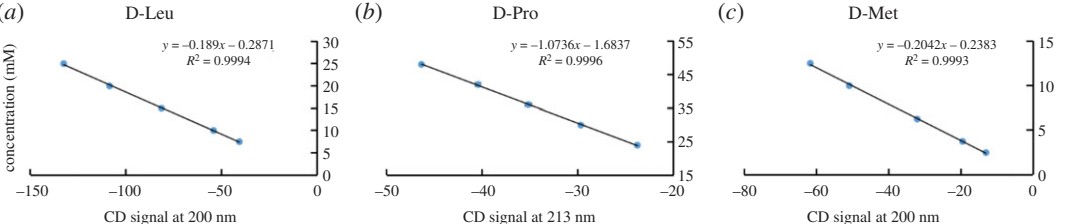

**Figure 3.** Calibration curves of D-amino acids using CD spectra. (*a*) Calibration curve of the D-Leu; (*b*) calibration curve of the D-Pro; (*c*) calibration curve of the D-Met. All the experiments data reported here were repeated three times.

**Table 2.** Calibration parameters for single amino acids in standard solution prepared by CD.

| analyte | linear range (mM) | linear equation (5 points) | correlation $R^2$ | LOD (S/N = 3) (mM) | LOQ (S/N = 10) (mM) |
|---|---|---|---|---|---|
| L-Leu | 5–25 | $y = 0.1927x - 0.53033$ | 0.9993 | 0.270 | 2.070 |
| L-Pro | 24–48 | $y = 1.0363x + 0.0023$ | 0.9998 | 0.820 | 2.730 |
| L-Met | 5–10 | $y = 0.1899x - 0.5837$ | 0.999 | 0.003 | 1.370 |

shown in table 2 and electronic supplementary material, table S2. The coefficient of determination ($R^2$) is at least 0.999 for calibration curves. The signal-to-noise ratio of 3 (S/N = 3) is the limit of detection (LOD) and the signal-to-noise ratio of 10 (S/N = 10) is the limit of quantification (LOQ).

Besides chiral aas, the quantitative analysis of chiral nucleoside and chiral quinine were also studied. These results showed good correlation between concentration and CD signal intensity (electronic supplementary material, figures S21–S22 and tables S3–S4).

This method has several features that make it particularly simple, rapid and universal to determine the concentration of single chiral aa. Other chiral analysis techniques (LC, MS, NMR and so on), which require chiral reagents or chiral columns, are more expensive and time-consuming.

## 3.2. Quantification of L- and D-amino acid in enantiomeric mixture

Subsequently, we wondered if single enantiomer concentration in mixtures of L- and D-aa could be determined using CD. To verify the general utilities of the above derivations, enantiomer mixtures were analysed with known concentration samples using CD along with LC (in this study, CD and LC were used independently, not LC-CD). As expected, the experimental values, which obtained through the linear equations (2.6) and (2.7) computing, matched well with the corresponding calculated concentration of L-aa and D-aa in the enantiomer mixtures (table 3 and figure 4; electronic supplementary material, figures S23–S31 and table S5). The LOD and concentration range of enantiomeric mixture are those of $c_a$ and $c_b$, where $c_a$ and $c_b$ are obtained by the calibration curve of CD signal ($|c_1 - c_2|$) and LC signal ($c_1 + c_2$), respectively. More specifically, the experimental values of Leu, Pro and Met were acceptably different from the calculated concentrations, and the differences in percentage were between 0 and 9.4. It indicates that the method using CD and LC for the determination of single enantiomer concentration in mixtures of L- and D-aa is acceptable.

## 3.3. Effect of pH on the CD signal of chiral amino acid

Finally, we wondered if pH does interfere with the CD signal of the aa. To test the pH applicable range for CD to determine the concentration of aa, three L-aa (10 mM L-Leu, 10 mM L-Pro and 5 mM L-Met) samples with known concentrations were studied in the range of pH1~12. The results demonstrated that although the CD signal intensity of aa at the same concentration had varied with different pH conditions, the signal intensity were very close to each other in the pH range from 3 to 8 (figure 5; electronic supplementary material, figure S32). Thus, the reproducibility of aa signal intensity using CD was accepted within a certain pH range around 7.

**Table 3.** Quantification of L-/D-amino acids in enantiomeric mixtures by CD and LC spectra. $\Delta$ diff.$=|Exp. - Cal|/Cal.*100\%$. All of difference are less than 10%. All the experiments data reported here were repeated three times.

| aa | | 60:40[b] | | 80:20[b] | | 20:80[b] | | 55:45[b] | |
|---|---|---|---|---|---|---|---|---|---|
| | | L-aa | D-aa | L-aa | D-aa | L-aa | D-aa | L-aa | D-aa |
| Leu (mM) | 8[a] | | | | | | | | |
| | Cal.[c] | 4.80 | 3.20 | 6.40 | 1.60 | 1.60 | 6.40 | 4.40 | 3.60 |
| | Exp.[d] | 4.80 | 3.50 | 6.40 | 1.60 | 1.60 | 6.50 | 4.40 | 3.80 |
| | $\Delta$ diff[e] | 0 | 9.4% | 0 | 0 | 0 | 1.6% | 0 | 5.6% |
| | 13[a] | | | | | | | | |
| | Cal.[c] | 7.80 | 5.20 | 10.40 | 2.60 | 2.60 | 10.40 | 7.15 | 5.85 |
| | Exp.[d] | 8.20 | 5.60 | 10.80 | 2.70 | 2.80 | 10.90 | 7.38 | 6.24 |
| | $\Delta$ diff[e] | 5.1% | 7.7% | 3.8% | 3.8% | 7.7% | 4.8% | 3.2% | 6.7% |
| | 18[a] | | | | | | | | |
| | Cal.[c] | 10.80 | 7.20 | 14.40 | 3.60 | 3.60 | 14.40 | 9.90 | 8.10 |
| | Exp.[d] | 11.60 | 7.70 | 14.90 | 3.50 | 3.50 | 14.80 | 10.50 | 8.70 |
| | $\Delta$ diff[e] | 7.4% | 6.9% | 3.5% | 2.8% | 2.8% | 2.8% | 6.1% | 7.4% |
| Pro (mM) | 25[a] | | | | | | | | |
| | Cal.[c] | 15.00 | 10.00 | 20.00 | 5.00 | 5.00 | 20.00 | 13.75 | 11.25 |
| | Exp.[d] | 15.00 | 10.00 | 20.00 | 5.00 | 5.00 | 20.00 | 14.11 | 11.18 |
| | $\Delta$ diff[e] | 0 | 0 | 0 | 0 | 0 | 0 | 2.6% | 0.6% |
| | 32[a] | | | | | | | | |
| | Cal.[c] | 19.20 | 12.80 | 25.60 | 6.40 | 6.40 | 25.60 | 17.60 | 14.40 |
| | Exp.[d] | 19.30 | 13.00 | 25.40 | 7.00 | 6.50 | 25.60 | 17.90 | 14.30 |
| | $\Delta$ diff[e] | 1.6% | 1.6% | 0.8% | 9.4% | 1.6% | 0 | 1.7% | 0.7% |
| | 40[a] | | | | | | | | |
| | Cal.[c] | 24.00 | 16.00 | 32.00 | 8.00 | 8.00 | 32.00 | 22.00 | 18.00 |
| | Exp.[d] | 24.00 | 16.00 | 31.00 | 8.00 | 8.00 | 32.00 | 22.00 | 18.00 |
| | $\Delta$ diff[e] | 0 | 0 | 3.1% | 0 | 0 | 0 | 0 | 0 |

(Continued.)

**Table 3.** (*Continued.*)

| aa | | 60 : 40[b] | | 80 : 20[b] | | 20 : 80[b] | | 55 : 45[b] | |
|---|---|---|---|---|---|---|---|---|---|
| | | L-aa | D-aa | L-aa | D-aa | L-aa | D-aa | L-aa | D-aa |
| Met (mM) | 7[a] | | | | | | | | |
| | Cal.[c] | 4.20 | 2.80 | 5.60 | 1.40 | 1.40 | 5.60 | 3.85 | 3.15 |
| | Exp.[d] | 4.30 | 2.70 | 5.40 | 1.50 | 1.50 | 5.50 | 3.73 | 3.28 |
| | Δ diff[e] | 2.4% | 3.6% | 3.6% | 7.1% | 7.1% | 1.8% | 3.1% | 4.1% |
| | 4[a] | | | | | | | | |
| | Cal.[c] | 2.40 | 1.60 | 3.20 | 0.80 | 0.80 | 3.20 | 2.20 | 1.80 |
| | Exp.[d] | 2.20 | 1.70 | 2.90 | 0.80 | 0.80 | 3.10 | 2.00 | 1.90 |
| | Δ diff[e] | 8.3% | 6.3% | 9.4% | 0 | 0 | 3.1% | 9.1% | 5.6% |
| | 6[a] | | | | | | | | |
| | Cal.[c] | 3.60 | 2.40 | 4.80 | 1.20 | 1.20 | 4.80 | 3.30 | 2.70 |
| | Exp.[d] | 3.50 | 2.50 | 4.60 | 1.30 | 1.20 | 4.80 | 3.20 | 2.80 |
| | Δ diff[e] | 2.8% | 4.2% | 4.2% | 8.3% | 0 | 0 | 3.0% | 3.7% |

[a]Concentration of L-aa and D-aa before mixing.

[b]The ratio of the mixture of L-aa and D-aa of the same concentration ($c_2$).

[c]The calculated value of the amino acid.

[d]The experimental value the amino acid.

[e]The relative error between the calculated value and the experimental value.

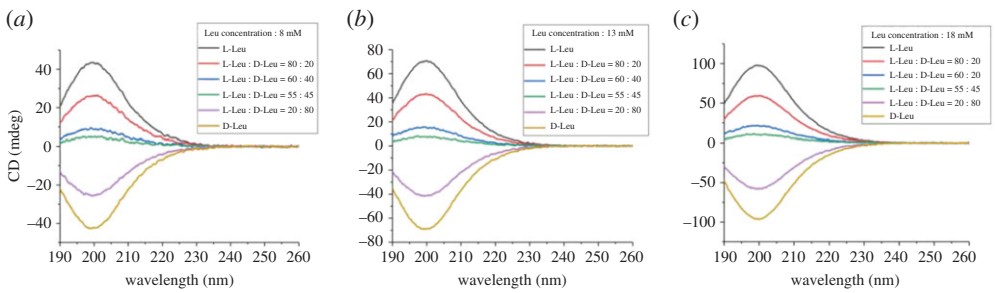

**Figure 4.** ECD spectra of Leu enantiomeric mixtures. (*a*) Various mixing ratio of 8 mM L-Leu and 8 mM D-Leu; (*b*) various mixing ratio of 13 mM L-Leu and 13 mM D-Leu; (*c*) various mixing ratio of 18 mM L-Leu and 18 mM D-Leu.

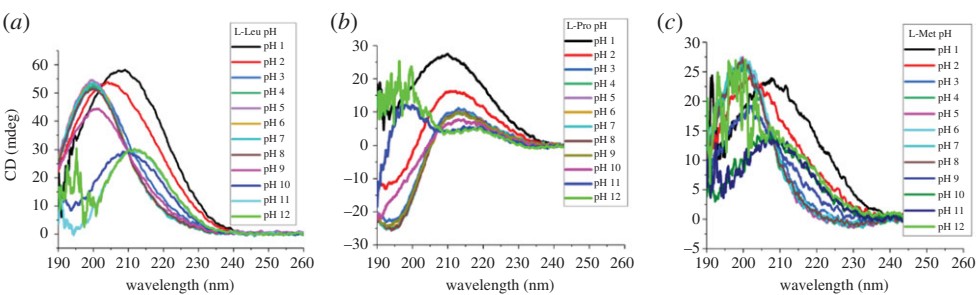

**Figure 5.** Effect of pH on the CD spectra of amino acids. (*a*) 10 mM L-Leu of various pH; (*b*) 10 mM L-Pro of various pH; (*c*) 5 mM L-Met of various pH. The results demonstrated that although the CD signal intensity of amino acid at the same concentration had varied with different pH conditions, the signal intensities were very close to each other in the pH range from 3 to 8. All the experiments data reported here were repeated three times.

# 4. Conclusion

In conclusion, we illustrated a new method for the quantification of L-/D-aa based on the calibration curves of the CD spectra. This method has several features that make it particularly simple, rapid and universal to determine the concentration of single chiral aa. Meanwhile, we can quantify the single enantiomer concentration in mixtures of L- and D-aa using calibration curves from the CD and LC spectra. More speculatively, although this method was limited to chiral aa in this study, it promising to be expanded for other chiral molecules (examples nucleoside and quinine of chirality in the §3.1 of this study). Overall, the present study has the potential to become a rapid, simple and general method for determination of chiral molecules concentration with broad applications, such as the chiral analysis of active pharmaceutical ingredient.

Data accessibility. The datasets supporting this article have been uploaded as part of the electronic supplementary material.

Authors' contributions. All authors contributed to the study design. R.D. recorded the spectra and analysed the data. J.Y. and Y.Z. designed the method and drafted the manuscript. All authors approved the final manuscript.

Competing interests. The authors have no competing interests.

Funding. This work was supported by the National Natural Science Foundation of China (grant nos. 91856126 and 42003062) and the Scientific Research Foundation of Graduate School of Ningbo University (grant no. IF2020156).

Acknowledgements. We are indebted to Dr Yile Wu and Dr Songsen Fu (Ningbo University) for many constructive comments on this manuscript.

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
