## [Peer Review File · Royal Society Open Science]

Review History

RSOS-201963.R0 (Original submission)

Review form: Reviewer 1

Is the manuscript scientifically sound in its present form?

No

Are the interpretations and conclusions justified by the results?

No

Is the language acceptable?

Yes

Do you have any ethical concerns with this paper?

No

Have you any concerns about statistical analyses in this paper?

No

Recommendation?

Major revision is needed (please make suggestions in comments)

Comments to the Author(s)

A novel analytical methods to quantify enatiopure amino acids as well as their enantiomeric mixtures have been developed by Ding et. al. with circular dichroism spectroscopy (and Liquid Chromatography). The method could supplement the currently used chiral HPLC analysis and is worth investigation. However, the reported work requires some major revisions before it could be published, some specific comments are provided below:

1, Chiral LC have been very mature in analyzing enatio-pure compounds or enantiomeric mixtures both qualitatively and quantitatively. Thus the claim " It is worth noting that, there is no general method for chiral analysis" by the authors might be over claimed. Still, the method developed by the authors could be a very promising one to complement the current ones.

2, The derivatizations of Equations 6 and 7 are not clear, especially equation 6. What do "nCD", "cCD", " nLC" as well as "cLD" stand for? Any difference between "nCD" and " nLC"; "cCD" and "cLD" ?

3, for section 4.1, " Quantification of single chiral amino acid" , what are the LOD (limit of detection) and linear range of the method for detection of Met, Pro and Leu?

4, in table 2, same question, what are the LOD (limit of detection) and linear range of the method for detection of Met, Pro and Leu enantiomeric mixtures? Since the method involves the calibration curves of both LC and CD (circular dichroism), the LOD and linear range should be carefully determined.

5, in table 2, the authors determined samples of Leu, Pro and Met in different concentrations and enantiomeric ratios, and compared their calculated values with experimental values. However, the circular dichroism spectral of the samples based on which the experimental values were determined were not provided! These spectral should be provided in the supplementary materials. Further, without the circular dichroism spectral , one could not tell whether it's a 60:40 or 40:60 enantiomeric ratio for the samples.

6, Last, the authors should go through the manuscript carefully and fix typos and grammar mistakes, to name a few:

page 1, line 31, " chiral molecule" should be "chiral molecules"

page 1, line 42, "liquid chromatographic" should be "liquid chromatography"

Review form: Reviewer 2

Is the manuscript scientifically sound in its present form?

No

Are the interpretations and conclusions justified by the results?

No

Is the language acceptable?

Yes

Do you have any ethical concerns with this paper?

No

Have you any concerns about statistical analyses in this paper?

No

Recommendation?

Major revision is needed (please make suggestions in comments)

Comments to the Author(s)

The Authors established a rapid quantitative analysis of single chiral amino acid in enantiomeric mixtures using CD and LC. It is a new and interesting approach for the analysis of chiral compounds. However, this manuscript is somewhat too simple, it should be revised a lot.

1. Summary: Which enantiomeric mixtures can be analyzed should be state
2. P4, Line 51, the concentration of amino acid should be stated.
3. The methodology of Quantification is quite incomplete, parameters such as specificity, accuracy, LOD, LOQ should be investigated.
4. Results and Discussion, the advantage of this method should be stated and this method should be compared with other existing methods for analyze chiral amino acids.

Decision letter (RSOS-201963.R0)

Dear Dr Ying:

Title: An electronic circular dichroism spectroscopy method for the quantification of L- and D-amino acids in enantiomeric mixtures

Manuscript ID: RSOS-201963

The editor assigned to your manuscript has now received comments from reviewers. We would like you to revise your paper in accordance with the referee and Subject Editor suggestions which can be found below (not including confidential reports to the Editor). Please note this decision does not guarantee eventual acceptance.

Please submit your revised paper before 05-Feb-2021. Please note that the revision deadline will expire at 00.00am on this date. If we do not hear from you within this time then it will be assumed that the paper has been withdrawn. In exceptional circumstances, extensions may be possible if agreed with the Editorial Office in advance. We do not allow multiple rounds of revision so we urge you to make every effort to fully address all of the comments at this stage. If deemed necessary by the Editors, your manuscript will be sent back to one or more of the original reviewers for assessment. If the original reviewers are not available we may invite new reviewers.

To revise your manuscript, log into <http://mc.manuscriptcentral.com/rsos> and enter your Author Centre, where you will find your manuscript title listed under "Manuscripts with Decisions." Under "Actions," click on "Create a Revision." Your manuscript number has been

appended to denote a revision. Revise your manuscript and upload a new version through your Author Centre.

RSC Associate Editor:
Comments to the Author:
(There are no comments.)

RSC Subject Editor:
Comments to the Author:
(There are no comments.)

Reviewers' Comments to Author:
Reviewer: 1

Comments to the Author(s)

A novel analytical methods to quantify enatiopure amino acids as well as their enantiomeric mixtures have been developed by Ding et. al. with circular dichroism spectroscopy (and Liquid Chromatography). The method could supplement the currently used chiral HPLC analysis and is worth investigation. However, the reported work requires some major revisions before it could be published, some specific comments are provided below:

1, Chiral LC have been very mature in analyzing enatio-pure compounds or enantiomeric mixtures both qualitatively and quantitatively. Thus the claim " It is worth noting that, there is no general method for chiral analysis" by the authors might be over claimed. Still, the method developed by the authors could be a very promising one to complement the current ones.

2, The derivatizations of Equations 6 and 7 are not clear, especially equation 6. What do “nCD”, “cCD”, “nLC” as well as “cLD” stand for? Any difference between “nCD” and “nLC”; “cCD” and “cLD”?

3, for section 4.1, “Quantification of single chiral amino acid”, what are the LOD (limit of detection) and linear range of the method for detection of Met, Pro and Leu?

4, in table 2, same question, what are the LOD (limit of detection) and linear range of the method for detection of Met, Pro and Leu enantiomeric mixtures? Since the method involves the calibration curves of both LC and CD (circular dichroism), the LOD and linear range should be carefully determined.

5, in table 2, the authors determined samples of Leu, Pro and Met in different concentrations and enantiomeric ratios, and compared their calculated values with experimental values. However, the circular dichroism spectral of the samples based on which the experimental values were determined were not provided! These spectral should be provided in the supplementary materials. Further, without the circular dichroism spectral, one could not tell whether it's a 60:40 or 40:60 enantiomeric ratio for the samples.

6, Last, the authors should go through the manuscript carefully and fix typos and grammar mistakes, to name a few:

page 1, line 31, “chiral molecule” should be “chiral molecules”

page 1, line 42, “liquid chromatographic” should be “liquid chromatography”

Reviewer: 2

Comments to the Author(s)

The Authors established a rapid quantitative analysis of single chiral amino acid in enantiomeric mixtures using CD and LC. It is a new and interesting approach for the analysis of chiral compounds. However, this manuscript is somewhat too simple, it should be revised a lot.

1. Summary: Which enantiomeric mixtures can be analyzed should be state

2. P4, Line 51, the concentration of amino acid should be stated.

3. The methodology of Quantification is quite incomplete, parameters such as specificity, accuracy, LOD, LOQ should be investigated.

4. Results and Discussion, the advantage of this method should be stated and this method should be compared with other existing methods for analyze chiral amino acids.

Author's Response to Decision Letter for (RSOS-201963.R0)

See Appendix A.

RSOS-201963.R1 (Revision)

Review form: Reviewer 2

Is the manuscript scientifically sound in its present form?

Yes

Are the interpretations and conclusions justified by the results?

Yes

Is the language acceptable?

Yes

Do you have any ethical concerns with this paper?

No

Have you any concerns about statistical analyses in this paper?

No

Recommendation?

Accept as is

Comments to the Author(s)

The authors had revised this manuscript carefully based on the review`s comments, I think it can be accepted now.

Decision letter (RSOS-201963.R1)

Dear Dr Ying:

Title: An electronic circular dichroism spectroscopy method for the quantification of L- and D-amino acids in enantiomeric mixtures

Manuscript ID: RSOS-201963.R1

It is a pleasure to accept your manuscript in its current form for publication in Royal Society Open Science. The chemistry content of Royal Society Open Science is published in collaboration with the Royal Society of Chemistry.

Yours sincerely,

Dr Laura Smith

Publishing Editor, Journals

RSC Associate Editor:
Comments to the Author:
(There are no comments.)

RSC Subject Editor:
Comments to the Author:
(There are no comments.)

Reviewer(s)' Comments to Author:
Reviewer: 2

Comments to the Author(s)
The authors had revised this manuscript carefully based on the review`s comments, I think it can be accepted now.

Appendix A

Thank you for your precious comments concerning my manuscript entitled “An electronic circular dichroism spectroscopy method for the quantification of L- and D-amino acids in enantiomeric mixtures” (ID: RSOS-201963). These comments are indeed valuable and very helpful for improving our manuscript. The comments also give important guidance for our research. We have carefully made relevant corrections based on your comments, which we hope meet with approval. The responses to the reviewers’ comments are as follows:

Comment #1-1: A novel analytical methods to quantify enatiopure amino acids as well as their enantiomeric mixtures have been developed by Ding et. al. with circular dichroism spectroscopy (and Liquid Chromatography). The method could supplement the currently used chiral HPLC analysis and is worth investigation.

Response: Thank you very much.

Comment #1-2: Chiral LC have been very mature in analyzing enatio-pure compounds or enantiomeric mixtures both qualitatively and quantitatively. Thus the claim “It is worth noting that, there is no general method for chiral analysis” by the authors might be over claimed. Still, the method developed by the authors could be a very promising one to complement the current ones.

Response: We are sorry for our negligence. It may be more appropriate to change the expression to “It is worth noting that although a general method to analyze chiral compounds base on chiral LC, other methods also display advantages and may complement the chiral analysis techniques”. We have made relevant modifications in the second paragraph of section 2 of our manuscript.

Comment #1-3: The derivatizations of Equations 6 and 7 are not clear, especially equation 6. What do “ n_{CD} ”, “ c_{CD} ”, “ n_{LC} ” as well as “ c_{LD} ” stand for? Any difference between “ n_{CD} ” and “ n_{LC} ”; “ c_{CD} ” and “ c_{LD} ”?

Response: Thank you for your reminder. We have made relevant modifications in the section 3.4 of our manuscript, specifically as follows:

$$n_a = |n_1 - n_2| \Rightarrow c_a V = |c_1 V - c_2 V| \Rightarrow c_a V = |c_1 - c_2| V \Rightarrow c_a = |c_1 - c_2| \quad (6)$$

Where n_1 and n_2 respectively are the amount of substance of the L-aa and D-aa, n_a is the absolute difference of n_1 and n_2 . c_1 and c_2 respectively are the concentration of the L-aa-and D-aa, c_a is the absolute difference of c_1 and c_2 . c_a is obtained by the calibration curve of CD signal.

$$n_b = n_1 + n_2 \Rightarrow c_b V = c_1 V + c_2 V \Rightarrow c_b V = (c_1 + c_2) V \Rightarrow c_b = c_1 + c_2 \quad (7)$$

Where n_1 and n_2 respectively are the amount of substance of the L-aa and D-aa, n_b is the sum of n_1 and n_2 . c_1 and c_2 respectively are the concentration of the L-aa-and D-aa, c_b is the sum of c_1 and c_2 . c_b is obtained by the calibration curve of LC signal.

Comment #1-4: for section 4.1, “Quantification of single chiral amino acid”, what are the LOD (limit of detection) and linear range of the method for detection of Met, Pro and Leu?

Response: Thank you for your attention. We have made relevant modification in the

Table 2 of section 4.1 of revised manuscript and Table S2 of section 3 of supporting information. The details of Table 2 are as follows:

Table 2. Calibration parameters for single amino acids in standard solution prepared by ECD

Analyte	Linear range (mM)	Linear equation (5 points)	Correlation, R^2	LOD (S/N=3) (mM)	LOQ (S/N=10) (mM)
L-Leu	5-25	$y=0.1927x-0.53033$	0.9993	0.270	2.070
L-Pro	24-48	$y=1.0363x+0.0023$	0.9998	0.820	2.730
L-Met	5-10	$y=0.1899x-0.5837$	0.999	0.003	1.370

The concentration range of standard solution analysis for each sample is shown. The coefficient of determination (R^2) is at least 0.999 for calibration curves. The signal-to-noise ratio of 3 (S/N=3) is the limit of detection (LOD) and the signal-to-noise ratio of 10 (S/N=10) is the limit of quantification (LOQ). (Analytical Methods, 2017. 9, 2840-2844.)

Comment #1-5: in table 2, same question, what are the LOD (limit of detection) and linear range of the method for detection of Met, Pro and Leu enantiomeric mixtures? Since the method involves the calibration curves of both LC and CD (circular dichroism), the LOD and linear range should be carefully determined.

Response: Thank you for your valuable comments on our manuscript and we really appreciate it. The limit of detection (LOD) and concentration range of enantiomeric mixture are those of c_a and c_b . Where c_a and c_b are obtained by the calibration curve of CD signal ($|c_1-c_2|$) and LC signal (c_1+c_2), respectively. We have made relevant modification in the Table 2 of section 4.1 and section 4.2 of revised manuscript and Table S2 of section 3 and Table S5 of section 6 of supporting information.

Comment #1-6: in table 2, the authors determined samples of Leu, Pro and Met in different concentrations and enantiomeric ratios, and compared their calculated values with experimental values. However, the circular dichroism spectral of the samples based on which the experimental values were determined were not provided! These spectral should be provided in the supplementary materials. Further, without the circular dichroism spectral, one could not tell whether it's a 60:40 or 40:60 enantiomeric ratio for the samples.

Response: We are sorry for our negligence. We have made relevant modification in the Fig. 4 of section 4.2 of revised manuscript and Fig. S23-S28 of section 5 of supporting information. The details of Fig. 4 are as follows:

Fig. 4. ECD spectra of Leu enantiomeric mixtures. a) Various mixing ratio of 8 mM L-Leu

and 8 mM D-Leu; b) Various mixing ratio of 13 mM L-Leu and 13 mM D-Leu; c) Various mixing ratio of 18 mM L-Leu and 18 mM D-Leu.

Comment #1-7: Last, the authors should go through the manuscript carefully and fix typos and grammar mistakes, to name a few:

page 1, line 31, “chiral molecule” should be “chiral molecules”

page 1, line 42, “liquid chromatographic” should be “liquid chromatography”

Response: Thank you for your reminder and we really appreciate it. We have made relevant modification in our manuscript.

Comment #2-1: The Authors established a rapid quantitative analysis of single chiral amino acid in enantiomeric mixtures using CD and LC. It is a new and interesting approach for the analysis of chiral compounds.

Response: Thank you very much.

Comment #2-2: Summary: Which enantiomeric mixtures can be analyzed should be state

Response: Thank you for your valuable comments on our manuscript and we really appreciate it. It may be more appropriate to change the expression to “With this study, we provide a new method for the chiral quantitative analysis of enantiomeric amino acids mixtures”. We have made relevant modifications in the summary of our manuscript.

Comment #2-3: P4, Line 51, the concentration of amino acid should be stated.

Response: Thank you for your reminder. We have made relevant modifications in the section 4.3 of revised manuscript.

Comment #2-4: The methodology of Quantification is quite incomplete, parameters such as specificity, accuracy, LOD, LOQ should be investigated.

Response: Thank you for your valuable comments on our manuscript and we really appreciate it. We have made relevant modifications in the Table 2 of revised manuscript and Table S2, S4 and S5 of supporting information. The details of Table 2 are as follows:

Table 2. Calibration parameters for single amino acids in standard solution prepared by ECD

Analyte	Linear range (mM)	Linear equation (5 points)	Correlation, R ²	LOD (S/N=3) (mM)	LOQ (S/N=10) (mM)
L-Leu	5-25	y=0.1927x-0.53033	0.9993	0.270	2.070
L-Pro	24-48	y=1.0363x+0.0023	0.9998	0.820	2.730
L-Met	5-10	y=0.1899x-0.5837	0.999	0.003	1.370

The concentration range of standard solution analysis for each sample is shown. The coefficient of determination (R²) is at least 0.999 for calibration curves. The signal-to-noise ratio of 3 (S/N=3) is the limit of detection (LOD) and the signal-to-noise ratio of 10 (S/N=10) is the limit of quantification (LOQ). (Analytical Methods, 2017. 9, 2840-2844.)

Comment #2-5: Results and Discussion, the advantage of this method should be stated and this method should be compared with other existing methods for analyze chiral amino acids.

Response: Thank you for your reminder. We are sorry for our negligence. We have made relevant modifications in the section 4.1 of our manuscript, specifically as follows:

This method has several features that make it particularly simple, rapid and universal to determine the concentration of single chiral amino acid. Other chiral analysis techniques (LC, MS, NMR and so on), which require chiral reagents or chiral columns, are more expensive and time-consuming.